# The Role of Electronic Noses in Phenotyping Patients with Chronic Obstructive Pulmonary Disease

**DOI:** 10.3390/bios10110171

**Published:** 2020-11-11

**Authors:** Simone Scarlata, Panaiotis Finamore, Martina Meszaros, Silvano Dragonieri, Andras Bikov

**Affiliations:** 1Unit of Internal Medicine, Campus Bio Medico University and Teaching Hospital, 00128 Rome, Italy; s.scarlata@unicampus.it; 2Unit of Geriatrics, Campus Bio Medico University and Teaching Hospital, 00128 Rome, Italy; p.finamore@unicampus.it; 3Department of Pulmonology and Sleep Disorders Centre, University Hospital Zürich, 8091 Zurich, Switzerland; martina.meszaros1015@gmail.com; 4Respiratory and Sleep Medicine Unit, Policlinico University Hospital, University of Bari Aldo Moro, 70124 Bari, Italy; sdragonieri@gmail.com; 5Wythenshawe Hospital, Manchester University NHS Foundation Trust, Manchester M23 9LT, UK; 6Division of Infection, Immunity and Respiratory Medicine, University of Manchester, Manchester M13 9NT, UK

**Keywords:** COPD, chronic obstructive pulmonary disease, e-nose, electronic nose, VOCs, volatile organic compounds

## Abstract

Chronic obstructive pulmonary disease (COPD) is a common progressive disorder of the respiratory system which is currently the third leading cause of death worldwide. Exhaled breath analysis is a non-invasive method to study lung diseases, and electronic noses have been extensively used in breath research. Studies with electronic noses have proved that the pattern of exhaled volatile organic compounds is different in COPD. More recent investigations have reported that electronic noses could potentially distinguish different endotypes (i.e., neutrophilic vs. eosinophilic) and are able to detect microorganisms in the airways responsible for exacerbations. This article will review the published literature on electronic noses and COPD and help in identifying methodological, physiological, and disease-related factors which could affect the results.

## 1. Chronic Obstructive Pulmonary Disease

Chronic obstructive pulmonary disease (COPD) is a common disorder of the respiratory system which is characterised by a progressive airflow limitation caused by exposure to noxious particles, usually tobacco smoke, in susceptible individuals [1]. However, other factors, such as premature birth, frequent childhood infections, asthma, or passive smoking, could also contribute to COPD [1]. The disease may affect the large airways, respiratory bronchioles, and lung parenchyma, however the extent of the involvement of different lung regions may vary [2] (Figure 1).

Large airway disease is characterised by mucus hypersecretion, ciliary and epithelial dysfunction, mucosal and submucosal inflammation, as well as enhanced bronchial blood flow. Patients may present with symptoms of chronic productive cough or chronic bronchitis. Most of these patients have small airway disease, which is characterised by airway inflammation, peribronchial fibrosis, and subsequent small airway narrowing. Parenchymal involvement is termed emphysema, and it is characterised by progressive loss of the lung tissue, impaired oxygen intake, and carbon dioxide removal. People with small airway disease and emphysema often complain of progressive shortness of breath. Although widely recognised as a progressive disease, the activity of disease varies largely between patients. Around half of patients have a rapid (≥50 mL/year loss) decline in forced expiratory volume in the first second (FEV_1_), a marker quantifying airway obstruction [3], and around 30% are prone to acute exacerbations, major events leading to health deterioration and associated with high healthcare burden and mortality [4].

COPD is diagnosed based on medical history, symptoms, and lung function showing fixed airflow obstruction. Although the diagnosis, especially the differential diagnosis from other lung diseases (i.e., asthma, bronchiectasis), is sometimes difficult, in most cases it can be made based on simple and cheap pulmonary function tests. It is important to have reliable biomarkers which could differentiate patients with eosinophilic airway inflammation and reflect on disease activity (i.e., predict lung function decline and future exacerbations). This is essential clinical information, as inhaled corticosteroids (ICS) seem to be more effective in patients with raised airway eosinophils [5], as well as patients with a high exacerbation burden [6]. On the other hand, in some patients recurrent exacerbations are maintained by colonising bacteria and patients may benefit from prophylactic antibiotic treatment [7]. Hence, biomarkers reflecting on bacterial colonisation and specifying bacteria would have significant clinical value. Similar to stable disease, acute exacerbations are also heterogeneous and patients may benefit from tailored treatment depending on the inflammatory profile [8] and infectious cause [9].

Exhaled breath analysis is a widely used technique for investigating airway diseases. It is totally harmless, and therefore can be performed even in very frail patients and during acute breathlessness, such as in exacerbation. Therefore, it has a great yet not fully explored clinical potential to distinguish patients with different inflammatory endotypes and airway microbiology. One of the most important limiting factors is the lack of standardisation [10] and the effect of various endogenous (airway calibre, comorbidities, etc.) and exogenous factors (diet, smoking, pollution) which may limit their use. Traditionally, techniques assessing breath biomarkers are divided into methods investigating volatile and non-volatile particles [10] and the measurement of breath temperature [11]. In this review, we will focus on the measurement of volatile organic compounds (VOCs) using electronic noses in COPD.

## 2. The Role of Electronic Noses in Breath Research

Exhaled breath contains thousands of VOCs usually in pico or nanomolar concentrations. The origin of these molecules is two-fold. Some of them are inhaled from the environment and exhaled later. Environmental molecules may also be absorbed in the human body, react with endogenous VOCs, or induce altered production of endogenous molecules. Another group of VOCs is released by the human body, including cells of the respiratory tract reflecting on their metabolism. The release of some of these endogenous volatile molecules is induced by inflammation and oxidative stress [12].

The precise detection and quantification of these molecules involves gas-chromatography mass-spectrometry (GC-MS). However, GC-MS is expensive, uses bulky equipment, and requires special expertise. Electronic noses are a relatively cheap and easy-to-use alternative to GC-MS [13]. These devices are composites of a sensor array and an in-built processor, and their function resembles that of biological olfaction receptors, as they are unspecific to single molecules and, upon activation by an odour, create a signal pattern. Electronic noses therefore cannot identify individual molecules but are able to compare and discriminate exhaled samples based on their molecular pattern, which is often called a “breathprint”.

Because individual VOCs generating a “breathprint” are not characterised during the analysis, it is essential to exclude environmental factors leading to altered VOC levels. As a first step, subjects are asked to avoid consuming food and beverages, not to smoke, and to refrain from physical exercise prior to breath collection. It is good practice if these events are recorded in a clinical research form together with the last medication taken. This may help in identifying outliers in the final analysis. Exhaled breath should be collected in a standardised way, either by a single expiratory method or a multiple breath technique, with custom-made or commercially available sampling devices [10]. The breath sample could be collected directly into disposable collection bags (e.g., polyethylene, Nalophan, Tedlar, Mylar, etc.) or canisters made of inert material or pre-concentrated into adsorbent cartridges (e.g., Breath Biopsy Cartridges, Tenax GR^®^) [14]. The stability of VOCs declines within hours and strongly depends on the material of the collection bags [15,16,17,18], therefore an immediate analysis is suggested following collection. The advantage of Tenax tubes is that they allow transportation and delayed measurement. Although storage and transportation in sorbent tubes significantly affected the “breathprint”, it did not influence the discrimination potential of an electronic nose to differentiate asthma and health [19]. A novel method has been introduced lately to analyse the “breathprint” directly during a single expiratory manoeuvre [20].

Breath collection is followed by the electronic nose analysis. Various electronic noses have been used in the field of breath research, including those based on conducting polymer sensor arrays (i.e., Cyranose 320) [21], metal oxides (i.e., Aeonose, Spironose) [22], nanomaterials [23], quartz microbalance (i.e., BIONOTE) [24], as well as colorimetric sensors [25]. All these devices are made of sensors able to interact with VOCs and generate electrical signals, however the mechanism underpinning the interaction is specific to the used array (e.g., piezoelectric effect for quartz microbalance electronic nose or oxidation for metal oxide devices). There is a difference between these sensor systems in terms of the sensitivity and selectivity to different VOCs, the stability of the sensor signal, and the effect of environmental factors such as temperature and humidity [14]. These differences need to be taken into account when comparing data originating from different sensor systems. Finally, sensor signals are integrated into a “breathprint”, which can be analysed with complex discrimination techniques. The exhaled breath sampling and analysis is summarised in Figure 2.

A common technical challenge which may affect the electronic nose measurements is the temporal drift of the sensors, which means that the steady state of the sensor responses changes over the time of usage. This drift can be divided into short- and long-term components. For conducting polymer sensors, the failure of the detachment of VOCs from the sensors or inability to regain the baseline conformation may contribute to short-term drift, which means that electronic nose measurements may influence the subsequent ones [26]. The oxidation of the polymers contributes to the long-term drift [27]. Exposing Cyranose 320, a commercially available electronic nose based on a conducting polymer sensor array, to methanethiol for 8 days resulted in a considerable sensor drift [28]. It was shown that even for VOCs with a low concentration, the calibration of the entire array response could be re-established robustly with the use of carefully chosen calibrants [21,29,30]. Another analytical method to normalise drift in complex sensor arrays could be orthogonal partial least squares normalisation [31]. A study by Bos et al. reported a significant sensor drift over 1 year, and the authors suggested transformation into standardised residuals by linear regression to normalise this [32]. The reproducibility of quartz microbalance sensors has been investigated in healthy subjects and COPD patients simultaneously by Incalzi et al. [33]. The authors found that the reproducibility was better in healthy subjects than in COPD, concluding that the variability of the disease, rather than sensor drift, is responsible for the variation in data [33]. Sensor drift has been noticed for metal oxide sensors as well [34]. Jaeschke et al. explored various statistical methods, including linear discriminant analysis (LDA), partial least squares discriminant analysis (PLS-DA), and direct orthogonalisation, and found that PLS-DA was particularly useful for correction [34]. The humidity of exhaled breath samples may contribute to sensor drift for gold nanoparticle-based sensor systems [34]. A compensation method based on relative humidity sensors has been proposed to address this issue [34]. Instrumental drift has been reported even for GC-MS [35]. Rodriguez-Perez et al. applied filtering and additive, multiplicative, and multivariate drift corrections and was able to improve the classification of COPD subjects based on exhaled breath analysis [35]. For a detailed description of the electronic nose technology in breath research, we refer to previously published review articles [13,14,21,23,27,36,37,38,39,40,41].

Another important aspect is the data analysis, as it involves complex pattern recognition and comparative statistical methods [42]. Not surprisingly, the results may be very different based on which method is used [43]. There is no single, recommended statistical method to be used in breath analyses [10], however the European Respiratory Society’s statement recommended careful pre-processing, normalisation, and environmental correction as part of the analysis as well as the validation of the results by an independent cohort [10]. Smolinska et al. summarised the normalisation and pre-processing techniques for exhaled breath analysis [44] with a detailed description of their advantages and drawbacks. Environmental contamination removal can be achieved with an analysis of the alveolar gradient by sampling the environment together with the actual breath sample [20].

## 3. Altered Production and Kinetics of Exhaled VOCs in COPD 

Patients with COPD are usually middle-aged and elderly, and they tend to lose skeletal muscle with the progression of their disease. Therefore, the exhaled levels of those volatile organic compounds which are affected by age and metabolism, such as isoprene, acetone, and alkanes [45,46] may be compromised not by the disease but by senescence. Isoprene and acetone are found in relatively high concentrations in exhaled breath compared to other VOCs, and therefore these can contribute to the unspecific E-nose pattern significantly. Having said this, the E-nose pattern was found to be related to age in previous studies [47,48]. However, other reports found no influence of age on electronic noses to discriminate people with obstructive airway disease from healthy controls [49,50]. Patients with COPD may have an altered lifestyle, including regular exercise and diet [51]. Indeed, a study by Gaida et al. reported that the differences in VOC profiles between two groups of COPD patients sampled in different German cities could be due to lifestyle-related factors [52], however environmental factors, such as air pollution, could also contribute [53]. A number of studies have reported altered exhaled VOC profile associated with the consumption of certain foods and beverages [54,55,56]. In addition, exercise could alter “breathprint”, likely due to increased metabolism and oxidative stress in the airways as well as increased cardiac output [57,58,59]. However, in patients with COPD, the liberation of VOCs may be compromised due to dynamic hyperinflation during exercise. A relationship between exercise tolerance and the profile of exhaled VOCs has been previously reported [60]. The effect of gender on the composition of the exhaled VOC profile is less evident [55,61,62]. As alterations in exhaled “breathprint” may be related to hormonal changes rather than biological gender [63,64], this has to be investigated in women with COPD separately, as in most cases the disease is diagnosed post-menopause.

The release of blood-borne volatile compounds into exhaled breath depends on the alveolo-capillary barrier, which may be destroyed in patients with emphysema. This factor may need to be taken into account when interpreting VOC results. Emphysema could also lead to significant ventilation-perfusion heterogeneity and the collapse potential of the small airways. Therefore, even during normal exhalation, VOCs originating from peripheral airways may entrap, which could contribute to their ultimate concentration in exhaled breath. In line with this, certain VOCs have been associated with the extent of emphysema and low diffusion capacity, as well as airflow limitation in patients with COPD [65]. A relationship between airway calibre and “breathprint” obtained with an electronic nose has been shown by some [47], but not all [66] studies. Focusing on COPD, a significant correlation was reported between the lung function and electronic nose results obtained by the quartz microbalance sensor array [33].

COPD is characterised by airflow limitation, which could influence the levels of exhaled VOCs, especially if they are collected with an uncontrolled single expiratory manoeuvre. In addition, some VOCs such as ethanol may be released from the bronchial circulation, and their diffusion to the airway lumen is hampered in bronchial thickening [67]. This is further complicated during acute exacerbations, which are characterised by mucus hypersecretion (leading to a longer diffusion time from the bronchial circulation) and an increase in bronchial circulation (causing an increased release of VOCs) [2]. Theoretically, the exhalation flow rate may influence the concentrations of those exhaled VOCs which are produced in the airways [47]. In line with this, the levels of exhaled VOCs such as acetone [68], ethanol [54], isoprene [69,70], and pentane [69], as well as the whole “breathprint” [20], were influenced by exhalation flow, and different exhalation flow rates affected the utility of the electronic nose to diagnose lung cancer [47]. However, it is unclear how this affects the electronic nose results in COPD. As ventilation heterogeneity—more particularly, the opening and closing of the peripheral airways—may show inter-breath variability, and because of the expiratory flow dependency of exhaled VOC levels, the volume-targeted multiple breath collection method may be preferable to the single-exhalation technique in COPD (Figure 3).

The liberation of VOCs into exhaled breath and their entrapment into the airway lining fluid (ALF) is strongly affected by the temperature and acidity of the ALF. For instance, a significant relationship was reported between the “breathprint” obtained by the electronic nose and the pH of the exhaled breath condensate [59]. As the bronchial temperature [11] and acidity [71] may be different in COPD, this effect needs to be considered when interpreting the VOC results.

Finally, the exhaled compounds may originate from within the gastrointestinal tract. In line with this, a previous study found that gastro-oesophageal reflux disease can affect the “breathprint” in COPD [72].

## 4. Exhaled VOCs in Relation to Inflammation and Microbiome in COPD

It is well known that the inflammatory response in COPD is largely neutrophilic, but can also involve eosinophils to a lesser extent [71,73]. The presence and activity of inflammatory cells differ among patients subgroups and may be related to diverse pathophysiological pathways within the disease [74,75]. Biomarkers which reflect the inflammatory endotype of patients with COPD can help in a more effective tailored therapy, since a better response to corticosteroids in COPD with a higher presence of eosinophils has been clearly shown [76,77,78].

Several studies have demonstrated that there is an association between exhaled VOCs and the presence of inflammatory cells in subjects affected by COPD [65,79,80,81]. Fens et al. showed that exhaled VOCs are associated with differential sputum cell counts and soluble sputum markers of activated neutrophils and eosinophils in mild and moderate COPD, suggesting the potential of “breathprint” as a non-invasive biomarker in relatively early stages of COPD [79]. Furthermore, studies with GC-MS reported numerous VOCs linked with sputum inflammatory cells [65,80]. In particular, Schleich et al. identified VOCs discriminating between eosinophil and neutrophil cell cultures, regardless of their activation status [80]. Interestingly, Basanta et al. detected other types of VOCs which were able to distinguish between patients with COPD and healthy controls as well as to discriminate among subgroups of clinical interest, such as smokers with COPD versus asymptomatic smokers, and COPD subjects with higher sputum eosinophils or those with frequent exacerbations [65]. Moreover, de Vries et al. analysed exhaled VOCs with an electronic nose, showing an adequate prediction of blood eosinophils and neutrophils count in a pooled cohort of patients with asthma and COPD, irrespective of their underlying disease [81]. These results are very similar to those in asthma, where the electronic nose response not only correlated with bronchoalveolar lavage eosinophils [82], but was able to discriminate the eosinophilic, neutrophilic, and paucigranulocytic sputum profile [83]. However, despite promising studies indicating a significant correlation between exhaled VOCs and the presence of inflammatory cells in blood or sputum, the validation of results is yet to be achieved.

Numerous studies have compared the exhaled VOCs of COPD patients during exacerbations and stable disease [84,85,86,87]. Using GC-MS, Pizzini et al. discriminated patients with acute COPD exacerbation from individuals with stable COPD as well as healthy controls. Following the exclusion of VOCs associated with smoking and those with high environmental concentration, they built a model based on a limited number of VOCs. The performance of this VOC pattern was superior to the C-reactive protein levels in discriminating stable and exacerbated patients [84]. Similarly, Gaugg et al. [85] showed that a profile of certain VOCs identified by GC-MS was different between frequent and non-frequent exacerbators at clinical stability, suggesting the potential of using breathomics for subphenotyping patients with COPD. Van Velzen et al. recruited patients with COPD at their stable state and followed them up at exacerbation and recovery. Breath samples were analysed with GC-MS and a platform of four electronic noses. The electronic nose platform was able to discriminate exacerbations from stable state as well as recovery with a cross-validated accuracy of 0.75, a sensitivity of 0.79, and a specificity of 0.71, without finding significant differences between baseline and recovery [86]. These results highlight the merit of VOC analysis in the follow-up of patients with COPD, as mild to moderate exacerbations are often difficult to diagnose due to the natural variability of the disease [1].

Due to compromised immune system as well as frequent steroid use, the airways of patients with COPD are often colonised by bacteria and fungi. These are associated with unique VOC patterns [88]. Using Cyranose 320 and LDA, Shafiek et al. demonstrated that the e-nose discriminates COPD exacerbation triggered by infections from non-infective exacerbations with an accuracy of 0.75 [87] and exacerbations from pneumonia with an accuracy 0.86–0.88, a sensitivity of 0.85–0.91, and a specificity of 0.86–0.75, depending on the presence of potentially pathogen microorganisms [87], allowing a timely and tailored antibiotic treatment. This accuracy is in line with another study where an electronic nose could discriminate viral and bacterial infection-induced exacerbation with an area under the receiver operating characteristic (ROC) curve (AUROC) of 0.74 [89]. Analysing the breath samples of 37 patients with COPD with the Cyranose 320 device, Sibila et al. concluded that patients with bacterial colonisation have a different “breathprint” [90]. More particularly, principal component analysis followed by ROC analysis showed an excellent classification (AUROC of 0.94). This study highlights the possibility of using electronic noses to identify patients who may benefit from a preventive antibiotic treatment.

## 5. The Effect of Smoking on Exhaled VOCs in COPD

Cigarette smoke is a mixture of more than 5000 chemical compounds, among which hundreds have toxic and/or carcinogenic effects [91]. A number of studies have shown that VOCs profiles are significantly influenced by smoking behaviour [49,52,53,92,93]. It must be recognised that the exhaled VOCs of current smokers can be influenced in two ways. First, VOCs may directly derive from the cigarette. In line with this, different exhaled compounds were identified and directly correlated with smoking status, smoking intensity, years of smoking, and depth of inhalation [94]. Moreover, strong differences were observed in the VOC composition of tobacco cigarette smokers and exhaled breath in the comparison with that of electronic cigarette smokers [95]. Such a direct influence on the exhaled VOC composition may be mitigated by asking patients to abstain from smoking for a certain period before sampling exhaled breath. It seems that a two-hour period may suffice to reduce the influence of smoking [38]. It is well known that cigarette smoke promotes oxidative stress in the human body by augmenting the number of free radicals [96,97]. This may result in a production of oxidative stress-related compounds which promote airway inflammation by activating neutrophils and eosinophils [98].

Very interestingly, electronic nose studies on patients with COPD showed no difference in VOCs profiles between current and ex-smokers [38,79,99], whereas in healthy controls, there were significant differences between current and non-smokers [38,93]. Similarly, a recent study by Rodriguez-Aguilar et al. showed no difference in the composition of exhaled breath obtained from patients with COPD who were divided into smoking-induced and household air pollution-induced groups [100]. Taken together, the aforementioned studies suggest that exhaled VOC-profile can be helpful in detecting COPD, irrespective from their smoking status. However, the difference between COPD and health has disappeared when COPD patients were compared to healthy, non-smoker volunteers [49]. It seems that smoking did not affect the discrimination potential of gold nanoparticles, carbon nanotube or colorimetric sensors [101,102,103].

Due to these discrepancies we suggest that smoking history should be taken into account when designing case-control studies.

## 6. The Effect of Medications on Exhaled VOCs in COPD

Pharmaceutical agents in COPD include bronchodilators, inhaled and systemic corticosteroids, roflumilast, antibiotics, mucolytics and theophylline [1]. Surprisingly, the number of studies investigating the effect of medications on exhaled VOC pattern is low.

Salbutamol is a short acting β2-receptor agonist. Gaugg and colleagues demonstrated that the levels of VOCs change after salbutamol inhalation in patients with chronic airway diseases (*n* = 13 asthma, 37 COPD), but not after inhaling placebo [104]. Similarly, Scarlata et al. reported a quantitative reduction in exhaled VOCs following inhaled bronchodilator treatment [105]. In line with this, the levels of exhaled VOCs were related to urinary salbutamol levels in patients with asthma [106]. Lazar et al. demonstrated a change in exhaled “breathprint” following salbutamol inhalation and the change was not due to alterations in the airway calibre, but the nebulised aerosols [66].

Corticosteroids are immunosuppressive drugs and are used in both stable (inhaled) and exacerbated (systemic) disease. Comparing ICS user and non-user patients with COPD, Fens et al. did not find a significant difference in the “breathprint” [49]. In line with this, analysing the effect of ICS on the potential of electronic noses to discriminate COPD from health, van Berkel did not find a significant influence [107]. Contrarily, Scarlata et al. reported a qualitative modification of exhaled VOC pattern in patients on ICS [105]. When investigating patients with severe asthma, a significant relationship between the exhaled “breathprint” and urine corticosteroid levels was found [106]. Analysing the same cohort of subjects, the stability of the electronic nose “breathprint” was determined by the systemic corticosteroid use [108].

It is likely that theophylline, antibiotics, roflumilast, and mucolytics may all effect exhaled VOCs, however these have not been investigated before.

## 7. The Effect of Respiratory and Non-Respiratory Comorbidities

Due to age, common aetiologies (i.e., smoking), and the direct effect of hypoxaemia and inflammation, COPD is often accompanied with comorbidities. These may individually affect the electronic nose results [14], but may also compromise electronic nose discrimination potential to detect COPD. For instance, obstructive sleep apnoea (OSA), which itself alters the levels of exhaled VOCs [109], also affects how precisely COPD can be detected with two different electronic noses [110,111]. Investigating 13 patients with OSA, 15 patients with COPD, and 13 patients with both diseases (overlap syndrome) with Cyranose 320 and LDA, Dragonieri et al. concluded that OSA was different from COPD patients and overlap syndrome, while there was no difference between the COPD and overlap groups [111]. In contrast, including patients with OSA in the control group, 25% of patients with COPD were inaccurately classified to the OSA group and 25% of the patients with OSA were incorrectly classified into the COPD group in the study by Scarlata et al., who used a quartz microbalance array and PLS-DA [110]. These two studies highlight that some components of the “breathprint” may not be disease-specific but may represent a common underlying mechanism (i.e., hypoxaemia). Likewise, the exhaled VOC levels may be different in cardiovascular diseases, such as heart failure and coronary artery disease [112], which frequently accompany COPD [113]. It is also true that COPD may compromise the diagnostic ability to detect other diseases, such as shown for lung cancer [114].

## 8. Electronic Nose Studies in COPD

As described above, the composition of exhaled VOCs in could be altered due to several endogenous and exogenous factors. This chapter summarises the published evidence for case-control studies (Table 1). First of all, it has to be emphasised that the electronic nose signal in COPD seems to be stable, with a within-day reproducibility of 0.80 and an overall mean between-day reproducibility around 0.70 [33,49,115].

Most studies have reported a good (AUROC or cross-validation value ≥0.70) to an excellent discrimination performance (≥0.90) of electronic noses in COPD regardless of comparator. However, it seems that this performance strongly depends on the number of control groups tested [110]. Notably, most of the studies conducted on multiple groups are generally underpowered with small sample size. For these reasons, targeting the optimal population represents a major issue deserving further investigation. An external validation set is highly recommended to strengthen the reliability of the results [10].

In most one-to-one comparison analyses, electronic noses have shown a good diagnostic ability, being able to discriminate COPD from healthy controls [49,87,90,100] and other chronic conditions carrying respiratory symptoms usually requiring differential diagnosis with COPD (i.e., asthma, obstructive sleep apnoea, lung cancer, chronic heart failure, etc.). Using Cyranose 320 and LDA, Fens et al. were able to discriminate patients with COPD from non-COPD smokers with a cross-validation value of 0.66. Interestingly, patients with COPD were not different from non-smoker controls [49]. In contrast, using Cyranose 320 and LDA as well, Sibila et al. could distinguish patients with COPD from non-COPD control subjects (77% ever-smoker) with a much higher cross-validation value (0.83-0.88) [90]. Using the same device and classification method, the study by Shafiek et al. reported an accuracy of 72% in classifying patients with stable COPD and healthy controls [87]. In a very recent study using the same device, adding support vector machines models to canonical discriminant analysis, a cross-validation value of 1.00 was achieved for the comparison between patients with COPD and healthy subjects [100]. COPD can be discriminated from chronic heart failure with an externally validated accuracy of 0.69, a sensitivity of 0.63, and a specificity of 0.74 [119], independently from age, smoking habit, and comorbidities, which have an impact on the VOC pattern [120]. Likewise, Fens and colleagues obtained an externally validated accuracy of 0.95, a sensitivity of 0.91, and a specificity of 0.94 in discriminating COPD smokers and former smokers from asthmatic patients with LDA [99]. Interestingly, the difference in VOC pattern between COPD and asthma cannot be attributed to the type (i.e., reversible or not) or degree of airways obstruction, because the externally validated discriminative accuracy remained almost the same [99]; these results suggest that COPD has a specific VOC pattern production, independent from the degree of airway obstruction. Regardless of smoking, COPD can be discriminated from OSA with an accuracy of 0.75-0.80 [110,111] and a sensitivity and specificity of 0.75, while the presence of both diseases in the same patient (i.e., overlap syndrome) cannot be clearly distinguished by COPD [111]. Likewise, COPD can be discriminated from lung cancer [20,116,121]. In all these studies, the participants performed exhaled breath analysis apart from spirometry and observed some restrictions in eating, smoking, and taking medication before the test, limiting its applicability in clinical practice. A combination of a metal-oxide semiconductor e-nose with a spirometer (i.e., “SpiroNose”, AMC, Amsterdam; Comon-Invent BV, Delft, The Netherlands) has represented a paramount step in the applicability of e-nose in clinical practice, allowing real-time analysis and eliminating the VOC collection and storage step. The study of De Vries and colleagues has demonstrated that SpiroNose is able to discriminate COPD Global Initiative for Obstructive Lung Disease (GOLD) stages II-IV from healthy controls, asthma, and lung cancer with a AUROC of 0.80, 0.81, and 0.88 [20], respectively, without the need for restrictions before the test.

Alpha-1 antitrypsin (AAT) deficiency is a relatively rare genetic cause for COPD. In a pilot study, an electronic nose was applied in the discrimination of 10 patients with AAT deficiency, 23 patients with COPD without AAT deficiency, and 10 healthy subjects. The authors concluded a good discriminative cross-validated accuracy based on LDA [118]. They also supplemented 11 AAT-deficient patients with human purified AAT and found a significant change in “breathprint”. This change could be either due to the direct effect of AAT on the exhaled VOC pattern or may represent immunological alterations due to the augmentation therapy [118].

The “breathprint” was associated with the exercise capacity of COPD patients, expressed by the six minute walking distance and the disease-specific prognostic index BODE (Body mass index, Obstruction, Dyspnea, and Exercise), and was be able to predict those patients with a steeper decline more accurately than GOLD classification with PLS-DA [60], helping clinicians tailor their interventions and follow up and also helping diagnose frail patients who could benefit from palliative care.

Although the technique is promising and is cheaper and easier to use than GC-MS, electronic noses are still more expensive than the current diagnostic spirometry and they warrant some expertise. In addition, due to the unspecific nature of the signals, they cannot easily be interpreted in clinical practice. Therefore, their role alone would be limited in diagnostic and differential diagnostic settings. However, their combination with traditional spirometry has merit in identifying endotypes and differentiating COPD from asthma with fixed airway obstruction [20,49,81]. Airway sampling using invasive techniques, such as bronchoscopy is not always feasible in COPD, and even sputum induction hold risks for patients with very severe COPD [122]. Although endotyping and monitoring airway inflammation hold essential clinical value [5], the currently used surrogates, such as blood eosinophils, only weakly correlate with their percentages in sputum [123]. In addition, it has recently been suggested that temporal variation, rather than the baseline values of blood eosinophilia, better predicted treatment response to inhaled corticosteroids in COPD [124]. The monitoring of airway inflammation via electronic nose holds clinical potential, and future studies should focus on this.

## 9. Summary

The composition of volatile organic compounds in exhaled breath may be different due to various endogenous (airway inflammation, altered kinetics) and environmental (smoking, medications) factors. This is reflected in the results of case-control studies comparing COPD to various control conditions. This review highlights the need for normalisation in technical (i.e., sensor drift, statistical methods), sampling-related (i.e., expiratory flow rate), and patient-related (i.e., diet, smoking, medication abstinence prior to sampling) factors to make the results more comparable. Although the technique is currently limited, we believe that, through the international standardisation of the method, this would be a valuable in the clinical assessment of the airway inflammation of patients with COPD.

## Figures and Tables

**Figure 1 biosensors-10-00171-f001:**
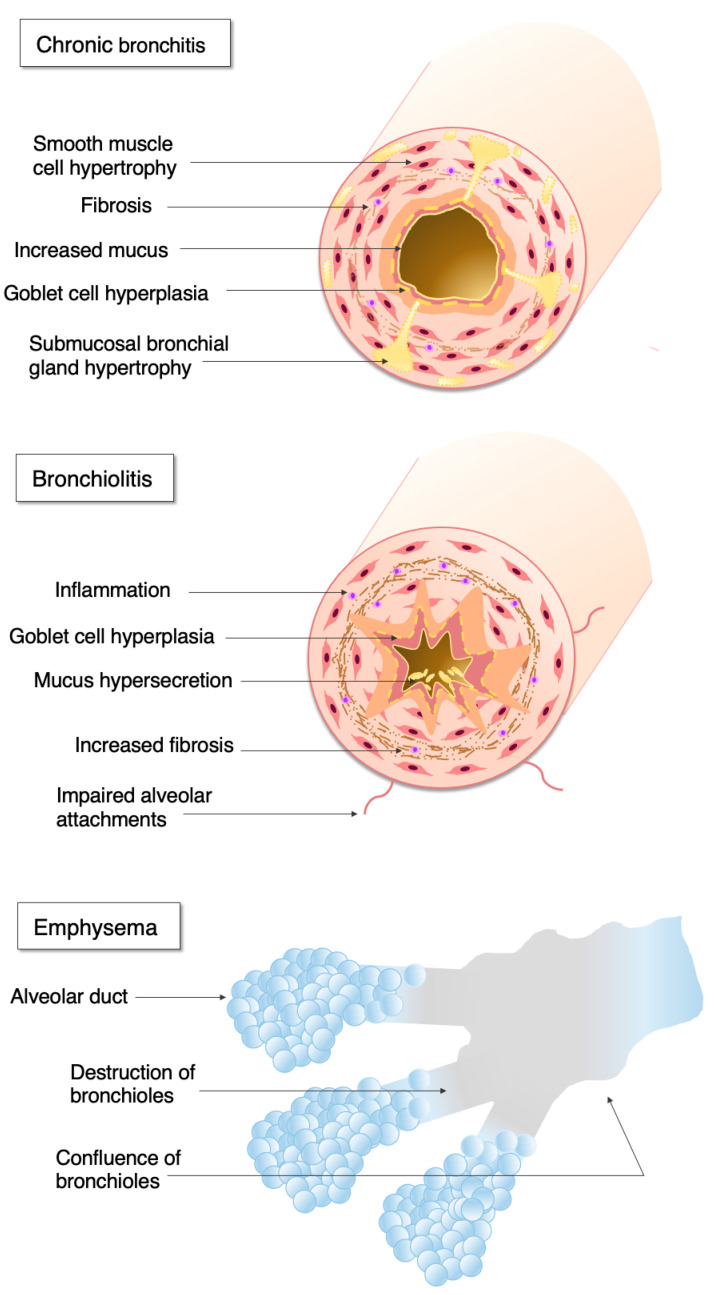
The pathophysiology of chronic obstructive pulmonary disease.

**Figure 2 biosensors-10-00171-f002:**
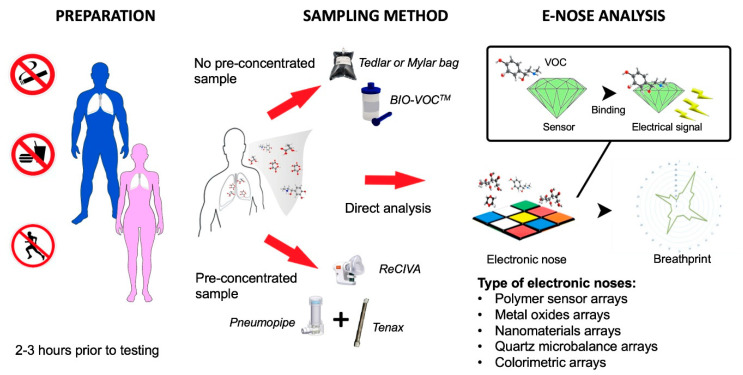
Algorithm for the exhaled breath collection and analysis with electronic nose.

**Figure 3 biosensors-10-00171-f003:**
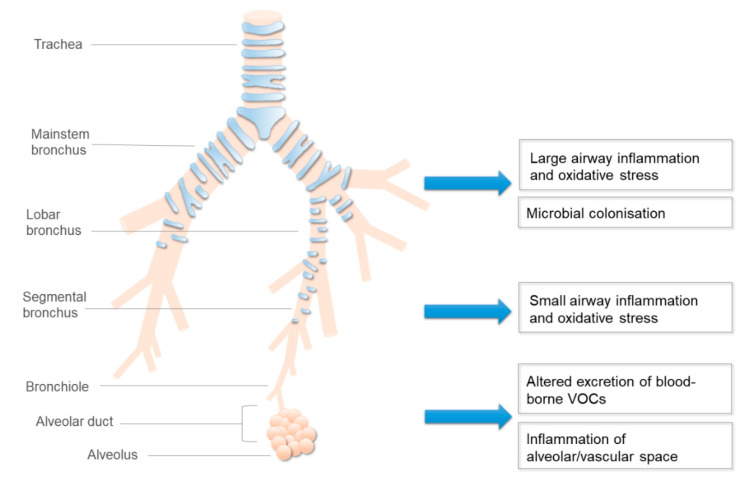
The mechanisms of altered production of exhaled volatile organic compounds (VOCs) in COPD.

**Table 1 biosensors-10-00171-t001:** Clinical studies conducted on electronic noses in patients with COPD.

Comparator Group	Device	Number of Subjects	Classification Technique	Sensitivity (%)	Specificity (%)	Cross-Validation Value (%)	Remarks	Reference
Healthy	Cyranose 320	N = 37 COPD N = 13 H	LDA	83	76	79	COPD vs. H	[90]
Infection	Cyranose 320	N = 74 ECOPD N = 19 ECOPD + P N = 50 COPD N = 30 H	LDA	72	67	ND	ECOPD vs. COPD	[87]
				88	75		ECOPD + P vs. COPD	
				91	75		ECOPD + P vs. ECOPD	
	Aeonose	N= 22 COPD + BI N = 21 COPD without BI N = 18 COPD + VI N = 25 COPD without VI	ANN	73	76	ND	COPD + VI vs. COPD without VI	[89]
				83	72		COPD + BI vs. COPD without BI	
Lung cancer	Cyranose 320	N = 10 LC N = 10 COPD N = 10 H	LDA	ND	ND	85	LC vs. COPD	[116]
						80	LC vs. H	
		N = 20 LC N = 31 COPD	ROC analysis based on principal components	80	48	ND	Diagnostic accuracy increased when combined with sputum hypermethylation	[117]
	Custom made colorimetric sensor	N = 18 COPD N = 49 LC N = 21 H N = 15 IPF N = 20 SR N = 20 PAH	Random forest method	73	72	ND	LC	[103]
Smoking	Cyranose 320	N = 88 COPD + S N = 28 COPD + HAP N = 178 H	LDA + SVM	100	97.8	100	COPD vs. H	[100]
				ND	98.1	100	COPD + S vs. H	
				ND	97.5	100	COPD + HAP vs. H	
				ND	2.5	75.7	COPD + S vs. COPD + HAP	
Asthma and lung cancer	SpiroNose	N = 31 COPD N = 37 A N = 31 LC N = 45 H	LDA	ND	ND	78	COPD vs. H	[20]
				ND	ND	81	COPD vs. A	
				ND	ND	80	COPD vs. LC	
				ND	ND	87	A vs. H	
				ND	ND	68	A vs. LC	
				ND	ND	88	LC vs. H	
Asthma and Smoking	Cyranose 320	N = 20 A N = 30 COPD N = 20 non-S N = 20 S	LDA	ND	ND	96	A vs. COPD	[49]
				ND	ND	95	A vs. non-S	
				ND	ND	93	A vs. S	
				ND	ND	66	COPD vs. S	
				ND	ND	NS	COPD vs. non-S	
Asthma	Cyranose 320	N = 40 COPD N = 60 A	LDA	85	90	88	COPD vs. fixed A (N = 21)	[99]
				91	90	83	COPD vs. reversible A (N = 39)	
	SpiroNose	N = 115 COPD N = 206 A	Not performed	ND	ND	NS	Five significant combined asthma and COPD clusters	[81]
OSA	Cyranose 320	N = 15 COPD N = 13 OSA N = 13 OVS.	LDA	ND	ND	96.2	OSA vs. OVS	[111]
				ND	ND	82.1	OSA vs. COPD	
				ND	ND	67.9	COPD vs. OVS	
	Custom made QMB	N = 20 COPD N = OSA + NH N = 20 OSA + H N = 20 O N = 56 H	PLS-DA	44	93	ND		[110]
Alpha 1-antitripsin deficiency	Cyranose 320	N = 10 COPD with AAT N = 23 COPD without AAT N = 10 H	LDA	ND	ND	58	AAT vs. non-AAT	[118]
				ND	ND	68	non-AAT vs. H	
				ND	ND	62	AAT vs. H	
Congestive heart failure	BIONOTE	N = 103 COPD N = 89 CHF N = 117 H	PLS-DA	80	82	ND	CHF vs. H	[119]
				63	74	ND	CHF vs. COPD	

A = asthma; AAT = alpha 1-antitripsin deficiency; ANN = artificial neural network; CHF = congestive heart failure; COPD = chronic obstructive pulmonary disease; COPD + BI = COPD with bacterial infection; COPD + HAP = COPD with household air pollution; COPD + S = COPD with smoking; COPD + VI = COPD with viral infection; ECOPD = exacerbation of COPD; ECOPD + P = exacerbation of COPD with pneumonia; H = healthy controls; IPF = idiopathic pulmonary fibrosis; LC = lung cancer; LDA = linear discriminant analysis; O = obese controls; OSA = obstructive sleep apnoea; OSA + H = hypoxic OSA; OSA + NH = non-hypoxic OSA; OVS = overlap syndrome; PAH = pulmonary arterial hypertension; PLS-DA = partial least square discriminant analysis; QMB = quartz microbalance; ROC = receiver operating characteristic; S = smoker; SR = sarcoidosis; SVM = support vector machines.

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
