# Peer review of "The Role of Electronic Noses in Phenotyping Patients with Chronic Obstructive Pulmonary Disease"

_biosensors, 2020, doi:10.3390/bios10110171_

Round 1

Reviewer 1 Report

Overall a very nice summary of COPD and VOCs with a well organized layout. The literature has been properly cited.

While the content is there, parts 1 and 2 require English language editing to improve the grammar. The subsequent sections require minor grammatical improvements.

Author Response

Comment: Overall a very nice summary of COPD and VOCs with a well-organized layout. The literature has been properly cited.

While the content is there, parts 1 and 2 require English language editing to improve the grammar. The subsequent sections require minor grammatical improvements.

Response: Thank you for your comments. The article has been reviewed by a native English speaker (Mr Rhys Tudge) and has been revised accordingly.

Reviewer 2 Report

The aim of the paper was to demonstrate the measurement of volatile organic compounds (VOCs), particularly, in case of COPD, using electronic noses. This is beneficial from both health and economical perspectives.

There are however a few things to take into consideration to improve the quality of the paper. Please find the detailed review below:

It will be interesting to see a general visual/schematic representation of the breath analysis with the e-nose. E.g how the breath sample is collected etc.

The explanation for FEV1 is missing (line 50). It would be important to mention what kind of lung function test it covers.

Lines 95-96: „For detailed description of electronic nose technology in breath research we refer to previously published review articles [13,14,16,19-26].” Several articles have been referenced here but there is no actual description of how these tests can be performed with electronic noses.

Authors wrote that using the e-nose involves complex pattern recognition and comparative statistical methods. In line 98-100, “Not surprisingly, the results may be very different based on which method is used [28] There is no single, recommended statistical method to be used in breath analyses”. It is necessary to discuss the importance of possible data pre-treatment methods (e.g. drift correction).

However, no statistical methods were mentioned in the cited studies. In certain instances, E.g line 192, an accuracy of 0.86-0.88 was mentioned but without a clear reference to the method that produced that accuracy: was it SIMCA, LDA, PLS etc?

Many studies were cited but there was very little discussion. E.g in line 189-190: “A prospective study by van Velzen et al [60] showed that exhaled VOC-profiles measured by both GC-MS and electronic nose differed during an exacerbation compared to clinical stability and recovery in a cohort of well-characterized patients with COPD”. How did they differ and what are the pros and cons from e-nose perspective? Similarly, also in line 248-249, it was written: “For instance, obstructive sleep apnoea, which itself alters the levels of exhaled VOCs [84] also affects how precisely COPD can be detected with two different electronic noses. How did the e-noses differ? What method was used for the detection? Etc. There are many of such statements that need to be properly discussed.

There is a column in Table 1 named “sensor system” but actually contains the name/type of e-noses that were used. Perhaps authors meant to include the actual sensor system of those e-noses. E.g conducting polymer sensor arrays, metal oxides, nanomaterials, quartz microbalance, colorimetric sensors etc. It will also be interesting to know what instruments were used in each of the cited studies; were they benchtop or handheld instruments?

Authors nicely discussed the influence of smoking and medication on breath analysis. How about diet? Is there no study about the effect of food on exhaled VOCs in COPD? Also, Drift is a major challenge that can influence the signal from sensor devices such as e-nose. It can be caused by diverse variations in environmental conditions and even the headspace generation time for e-nose. Do you think drift can influence breath analysis? Did any of the cited studies encounter this issue and how was it resolved?

The authors discuss different experimental subjects, e.g. smokers and non-smokers. Is information available in the literature on the effects of gender and lifestyle (e. g. regular sport activity) on the illness and diagnosability by e-nose?

The referenced study in lines 301-304 requires further clarification, the authors indicate the importance of “augmentation therapy” on the pattern of VOC, maybe explaining such effect in more details could help the readers understand more the purpose of such study.

The reasoning in lines 313-314 is rather out of place, as you wanted to prove just how suitable the instrument is for diagnostics: „Although the technique is promising, electronic noses are more expensive than the current diagnostic spirometry and they warrant some expertise.” The statement is inconsistent with line 85. This should be further discussed.

The paper lacks a conclusion. If the summary section is meant to be the conclusion, then, it should be devoid of citations and should include only the novelties from the authors perspective with respect to the aim of the paper.

When inserting abbreviations for the first time, the authors should state what they stand for, such as in lines 50 (FEV) and 188 (ROC-AUC).

The whole manuscript should be thoroughly checked for grammar, punctuations and tenses.

Please find some additional suggestions marked in the attached manuscript file.

Author Response

Reviewer 2

Comment: It will be interesting to see a general visual/schematic representation of the breath analysis with the e-nose. E.g how the breath sample is collected etc.

Response: Thank you for your suggestion. We added this figure to the manuscript.

Comment: The explanation for FEV1 is missing (line 50). It would be important to mention what kind of lung function test it covers.

Response: Thank you. This has now been clarified in the revised manuscript.

Comment: Lines 95-96: „For detailed description of electronic nose technology in breath research we refer to previously published review articles [13,14,16,19-26].” Several articles have been referenced here but there is no actual description of how these tests can be performed with electronic noses.

Response: Thank you for your comment. We expanded section 2 with a detailed description of sampling and analysis.

Comment: Authors wrote that using the e-nose involves complex pattern recognition and comparative statistical methods. In line 98-100, “Not surprisingly, the results may be very different based on which method is used [28] There is no single, recommended statistical method to be used in breath analyses”. It is necessary to discuss the importance of possible data pre-treatment methods (e.g. drift correction). However, no statistical methods were mentioned in the cited studies. In certain instances, E.g line 192, an accuracy of 0.86-0.88 was mentioned but without a clear reference to the method that produced that accuracy: was it SIMCA, LDA, PLS etc?

Response: Thank you for your comment. We expanded Section 2 by discussing the importance of pre-processing and normalisation as well as sensor drift. We also added information on which statistical methods were used in specific studies and updated Table 1 with this information.

Comment: Many studies were cited but there was very little discussion. E.g in line 189-190: “A prospective study by van Velzen et al [60] showed that exhaled VOC-profiles measured by both GC-MS and electronic nose differed during an exacerbation compared to clinical stability and recovery in a cohort of well-characterized patients with COPD”. How did they differ and what are the pros and cons from e-nose perspective? Similarly, also in line 248-249, it was written: “For instance, obstructive sleep apnoea, which itself alters the levels of exhaled VOCs [84] also affects how precisely COPD can be detected with two different electronic noses. How did the e-noses differ? What method was used for the detection? Etc. There are many of such statements that need to be properly discussed.

Response: We critically revised the manuscript and added more information when describing the individual studies.

Comment: There is a column in Table 1 named “sensor system” but actually contains the name/type of e-noses that were used. Perhaps authors meant to include the actual sensor system of those e-noses. E.g conducting polymer sensor arrays, metal oxides, nanomaterials, quartz microbalance, colorimetric sensors etc. It will also be interesting to know what instruments were used in each of the cited studies; were they benchtop or handheld instruments?

Response: We updated Table 1 with the information, if the E-noses were custom made or commercially available. We clarified the type of sensors used in the commercially available devices in section 2. The dimensions of the instruments were not always available in the original studies, therefore we did not add information if they were benchtop or handheld instruments.

Comment: Authors nicely discussed the influence of smoking and medication on breath analysis. How about diet? Is there no study about the effect of food on exhaled VOCs in COPD? Also, Drift is a major challenge that can influence the signal from sensor devices such as e-nose. It can be caused by diverse variations in environmental conditions and even the headspace generation time for e-nose. Do you think drift can influence breath analysis? Did any of the cited studies encounter this issue and how was it resolved?

Response: Thank you for your comments. The effect of diet has now been discussed in Section 3. We expanded section 2 by describing sensor drift and how this can be adjusted in electronic nose studies.

Comment: The authors discuss different experimental subjects, e.g. smokers and non-smokers. Is information available in the literature on the effects of gender and lifestyle (e. g. regular sport activity) on the illness and diagnosability by e-nose?

Response: Section 3 has been expanded to contain this information.

Comment: The referenced study in lines 301-304 requires further clarification, the authors indicate the importance of “augmentation therapy” on the pattern of VOC, maybe explaining such effect in more details could help the readers understand more the purpose of such study.

Response: Thank you for your comment. We described the referenced study in detail.

Comment: The reasoning in lines 313-314 is rather out of place, as you wanted to prove just how suitable the instrument is for diagnostics: „Although the technique is promising, electronic noses are more expensive than the current diagnostic spirometry and they warrant some expertise.” The statement is inconsistent with line 85. This should be further discussed.

Response: We do not think that there is a discrepancy between the two sentences. In line 85 we compared the price to GC-MS, in lines 313-314 to spirometry. We clarified this in the revised manuscript.

Comment: The paper lacks a conclusion. If the summary section is meant to be the conclusion, then, it should be devoid of citations and should include only the novelties from the authors perspective with respect to the aim of the paper.

Response: Thank you. Following your comment, the summary has been rewritten concluding the results described in the main text.

Comment: When inserting abbreviations for the first time, the authors should state what they stand for, such as in lines 50 (FEV) and 188 (ROC-AUC).

Response: Sorry for the mistake, this has been now corrected.

Comment: The whole manuscript should be thoroughly checked for grammar, punctuations and tenses.

Response: Thank you for your comments. The article has been reviewed by a native English speaker (Mr Rhys Tudge) and has been revised accordingly.

Comment: Please find some additional suggestions marked in the attached manuscript file.

Response: We appreciate your suggestions for corrections. These were implicated in the revised manuscript.

Reviewer 3 Report

The review is written quite well and comprehensive. 

Author Response

Reviewer 3

Comment: The review is written quite well and comprehensive.

Response: Thank you very much for your comment.

Round 2

Reviewer 2 Report

The authors corrected their manuscript based on reviewers' recommendations and it is of high quality to be accepted for publication.